# The Multifaceted Regulation of SnRK2 Kinases

**DOI:** 10.3390/cells10092180

**Published:** 2021-08-24

**Authors:** Justyna Maszkowska, Katarzyna Patrycja Szymańska, Adrian Kasztelan, Ewa Krzywińska, Olga Sztatelman, Grażyna Dobrowolska

**Affiliations:** 1Institute of Biochemistry and Biophysics, Polish Academy of Sciences, Pawińskiego 5a, 02-106 Warsaw, Poland; j.maszkowska@ibb.waw.pl (J.M.); akasztelan@ibb.waw.pl (A.K.); e.krzywinska@ibb.waw.pl (E.K.); 2Chair of Drug and Cosmetics Biotechnology, Faculty of Chemistry, Warsaw University of Technology, ul. Noakowskiego 3, 00-664 Warsaw, Poland; Katarzyna.Szymanska@pw.edu.pl

**Keywords:** SnRK2, protein phosphorylation, protein de-phosphorylation, osmotic stress, salinity, plant, cell signaling, signal transduction, ABA signaling

## Abstract

SNF1-related kinases 2 (SnRK2s) are central regulators of plant responses to environmental cues simultaneously playing a pivotal role in the plant development and growth in favorable conditions. They are activated in response to osmotic stress and some of them also to abscisic acid (ABA), the latter being key in ABA signaling. The SnRK2s can be viewed as molecular switches between growth and stress response; therefore, their activity is tightly regulated; needed only for a short time to trigger the response, it has to be induced transiently and otherwise kept at a very low level. This implies a strict and multifaceted control of SnRK2s in plant cells. Despite emerging new information concerning the regulation of SnRK2s, especially those involved in ABA signaling, a lot remains to be uncovered, the regulation of SnRK2s in an ABA-independent manner being particularly understudied. Here, we present an overview of available data, discuss some controversial issues, and provide our perspective on SnRK2 regulation.

## 1. Introduction

Plants growing in natural conditions are exposed to a continuously changing environment. To survive, they have had to develop numerous adaptation mechanisms. Upon recognition of an environmental change, plants trigger acclimatization signaling pathways in which protein kinases and phosphatases play crucial roles. This review focuses on SnRK2s—the enzymes engaged in plant development and growth as well as in the responses to a plethora of environmental stresses. SnRK2s together with several other signaling proteins, including other kinases, play a key role in switching between growth and stress response programs, allowing the plant to adjust to environmental changes [1,2].

An involvement of SnRK2s in abiotic stress signal transduction in plants was first recognized at the turn of the centuries [3,4,5,6,7]. Twenty years of studies on SnRK2s have demonstrated that they are present in every plant species studied, from algae and mosses to higher plants, and provided ample data describing their role in the plant defense against harsh environments (e.g., drought, salinity, pathogen infection, stress induced by heavy metal ions) (for review, see [8,9,10,11]). Basing on their phylogeny, SnRK2s have been divided into three groups. This classification correlates well with their sensitivity to ABA: group 1 are kinases non-responsive to ABA (in Arabidopsis SnRK2.1, 2.4, 2.5, 2.9 and 2.10), group 3—kinases strongly activated in response to ABA (in Arabidopsis SnRK2.2, 2.3 and 2.6), while group 2 comprises those not activated or weakly activated by ABA (in Arabidopsis SnRK2.7 and 2.8). In fact, group 2 kinases from rice are not activated by ABA and those from Arabidopsis are weakly activated by ABA, only when overexpressed in protoplasts or T87 cells [12,13,14]. Native SnRK2.7 and SnRK2.8 seem not to be activated by this hormone in plants, since no kinase activity was detected in extracts from ABA-treated Arabidopsis mutant plants lacking the group 3 (ABA-responsive) kinases [15].

Evolutionary studies indicate that group 3 is the ancient form of land plant SnRK2s, whereas group 1 is the most recent [16,17,18]; for a recent review, see [10]. It has been suggested that the precursor of SnRK2 genes (group 3-type) arose before the evolution of land plants and further duplication and sub-functionalization of those genes results in the formation of groups 1 and 2 after the separation of the evolutionary lineages, bryophytes and vascular plants.

SnRK2 kinases are comprised of a well-conserved N-terminal catalytic domain, similar to those of yeast sucrose non-fermenting 1 (SNF1) and animal AMP-activated protein kinase (AMPK), and a regulatory C-terminal domain consisting of two subdomains: domain I and domain II. Domain I of about 30 amino acids just after the kinase domain, also known as the SnRK2 box, is conserved in all SnRK2s and is required for ABA-independent activation in response to osmotic stress (Figure 1A). Domain II, comprising about 40 amino acids following domain I, is overall highly acidic with stretches of acidic amino acids, Glu (group 1) or Asp (groups 2 and 3) [12,14]. In the ABA-responsive kinases, it is also known as the ABA box, since it is required for the ABA-dependent activation [12,19].

All Arabidopsis SnRK2s, except SnRK2.9, and all rice SnRK2s (called SAPKs—stress/ABA-activated protein kinases) are activated in response to osmotic stress, in nature resulting from drought or salinity [13,14]. Consistent with this, an Arabidopsis decuple mutant lacking all SnRK2s grew poorly under osmotic stress, revealing the importance of SnRK2s in coping with the stress [20].

The activation of SnRK2s by osmotic stress is rapid and transient, suggesting their involvement in the early response. SnRK2.4 and SnRK2.10 (ABA-non-responsive SnRK2s) are activated in Arabidopsis roots already after 30 s of salt treatment and the activity disappears about 1 h later [21]. The activity of NtOSAK (*Nicotiana tabacum* Osmotic Stress Activated Kinase), a homologue of SnRK2.4 and SnRK2.10, is observed already after one minute of treatment of BY-2 cells with NaCl and lasts for approximately 2 h [7]. A fast and transient activation of SnRK2s from group 3 is induced also by ABA [13,22]. The activation by hyperosmotic stress and ABA occurs independently, since SnRK2s become activated in response to osmotic stress even in the ABA-deficient *aba1* mutant [23].

In non-stress conditions, SnRK2s are dispersed in both the cytoplasm and nucleus; an exception is the exclusively cytoplasmatic SnRK2.10. Upon salt or osmotic stress, some group 1 SnRK2s, but not ABA-responsive ones, relocalize to punctate structures in the cytoplasm [21,24,25], which for SnRK2.1 and SnRK2.4 were identified as P-bodies [25]. Additionally, SnRK2.4 was observed in membrane compartments [21]. The mechanism of those translocations has not been uncovered. It is also well known that group 2 and 3 SnRK2s phosphorylate substrates localized in the nucleus, cytoplasm, and cellular membrane.

Using various approaches, including phosphoproteomics, a large number of cellular targets of ABA-responsive SnRK2s have been identified, mostly proteins involved in stress responses. Their activity, stability, and/or localization depends on phosphorylation, moreover, quite often the changes evoked by phosphorylation/dephosphorylation are transient, which is in line with the transient nature of SnRK2 activity. The substrates of ABA-responsive SnRK2s include: transcription factors involved in expression of stress-responsive genes [26,27,28,29,30,31,32], ion channels regulating stomatal movements in an ABA-dependent manner [33,34,35,36], RNA-binding proteins, e.g., Hyponastic Leaves 1 and SERRATE, playing a crucial role in miRNA biogenesis [37], aquaporin (plasma membrane intrinsic protein 2;1) [38], proteins involved in flowering regulation [30], and many others. In contrast, only a few targets of group 1 SnRK2s are known so far: VARICOSE (VCS) and VCS-related protein involved in mRNA decay [25,39], and dehydrins Early Responsive to Dehydration 10 (ERD10) and ERD14 engaged in protecting proteins, lipids, and nucleic acids against adverse factors [40]. Comprehensive reviews on SnRK2s describing their targets have been published recently [9,11,41,42].

The transient nature of SnRK2s activation indicates that specific mechanisms responsible for both the activation and inactivation exist in plant cells to ensure adequate responses to environmental and developmental cues. SnRK2s are regulated both directly (by posttranslational modifications, protein–protein interactions, and alterations of their abundance) and indirectly, by modulation of the SnRK2-controlling proteins. The progress in the understanding of the regulation of ABA-activated SnRK2s and ABA signaling is enormous with numerous reports published each year. In contrast, the ABA-non-responsive SnRK2s have received much less attention; their role and the ABA-independent regulation of SnRK2s are still poorly understood, even though they are activated extremely rapidly in response to stress and therefore are apparently important in the early response. In this review, we try to cover most of the mechanisms and factors affecting SnRK2 activity and, as a consequence, the SnRK2 signaling pathways. By providing a comparison between the ABA-dependent and ABA-independent regulation of SnRK2s, we will highlight some major gaps in the understanding of these processes and indicate perspectives for future studies.

## 2. Mechanisms of SnRK2 Activation

Protein phosphorylation, which is a reversible modification, plays a key role in the regulation of SnRK2 activity. Although there are some differences in the activation/inactivation mechanisms between the ABA-responsive and non-responsive SnRK2s, the phosphorylation within their activation loops is mandatory for activity (Figure 1 and Figure 2). The residues whose phosphorylation is required for activity have been identified in SnRK2s from various species: *N. tabacum* NtOSAK (Ser158), Arabidopsis SnRK2.3 (Ser176), SnRK2.6 (Ser175) and SnRK2.10 (Ser158), and rice SAPK1 (Ser158) [14,19,23,43]. Some additional residues in the activation loops were also found to be phosphorylated in response to ABA and/or osmotic stress, such as Ser171 in SnRK2.6 and Ser154 in SnRK2.10 and NtOSAK. Although these residues are not crucial for activity (in vitro studies using recombinant proteins), their phosphorylation is required for SnRK2 activation *in planta* [43,44]. Phosphorylation of some other residues has an impact on SnRK2 activity as well [19,45].

### 2.1. Autophosphorylation and Self-Activation

SnRK2s activation through an autophosphorylation mechanism has been suggested since the residues crucial for activity were phosphorylated also in SnRK2s produced in *Escherichia coli* [46]. Crystallographic and biochemical studies on ABA-responsive kinases [47,48,49] provided a possible mechanism of SnRK2 self-activation (Figure 1B,C). Structural data indicated that the C-terminal regulatory SnRK2 box forms an α-helix lying in parallel against αC helix in the N-terminal lobe and stabilizing its position. The position of αC helix regulates the accessibility of the catalytic cleft. The “closed” conformation is characteristic for active kinases, while the “open” one corresponds to inactive kinases [50]. As unphosphorylated ABA-responsive SnRK2s exhibit basal activity, it is plausible that they can spontaneously switch between the “open” and “closed” conformations, which allows subsequent intermolecular phosphorylation and their full activation [47]. Arabidopsis SnRK2s differ in the efficiency of autophosphorylation; for example, SnRK2.6 exhibits much higher autophosphorylation than SnRK2.2 or SnRK2.3, suggesting that at least some SnRK2s could be additionally phosphorylated and activated by upstream kinases. A similar structure and autophosphorylation ability have been found for other SnRK2s, e.g., rice SAPK9 and sugarcane ScSAPK10 [51,52]. The activation by autophosphorylation fits well the model of activation of group 3 kinases in response to ABA in higher plants [46,47]. Briefly, when a plant grows in optimal conditions and ABA level is low, clade A type 2C protein phosphatases (PP2Cs), for instance ABA INSENSITIVE 1 and 2 (ABI1, ABI2), HYPERSENSITIVE TO ABA1 (HAB1) or ABA-HYPERSENSITIVE GERMINATION 3 (AHG3/PP2CA) interact with, dephosphorylate, and thereby inactivate group 3 SnRK2s. This changes when ABA level increases (e.g., in response to drought or salinity). PYRABACTIN RESISTANCE 1/PYR1-LIKE/REGULATORY COMPONENTS OF ABA RECEPTOR ABA receptors (hereafter referred to as PYLs) bind the hormone and form a complex with PP2Cs (their co-receptors). Ng et al. [47] suggested a two-step activation mechanism of ABA-responsive SnRK2s. Once a PP2C interacts with the ABA receptor, the phosphatase is inactivated due to a conformational change and the SnRK2 is released from inhibition. At this stage, SnRK2s are not phosphorylated but, according to conformational studies, they could stay partially active, which enables them to autophosphorylate and become fully active (Figure 1). Recent studies showed that autophosphorylation of SnRK2.6 (or phosphorylation by an upstream kinase) at several residues other than Ser175 (i.e., Tyr182, Tyr163, Ser164, Ser166, Ser167, and Ser267) is needed for the full activation [45,53]. Importantly, most of those residues are conserved both in the ABA-responsive SnRK2s and in the non-responsive ones. The ability to reconstruct an ABA signaling pathway using recombinant RCAR/PYR1/PYL ABA receptors, PP2Cs, and SnRK2s [46,54] indicated that indeed SnRK2s might be activated by autophosphorylation in response to ABA.

### 2.2. Phosphorylation by Raf-Like Kinases

Even though the proposed self-activation mechanism of ABA-responsive SnRK2s in response to ABA was very convincing, a complementary one depending on phosphorylation by upstream kinases was also considered. The best argument that SnRK2s can indeed be phosphorylated and activated by an upstream kinase(s) came from studies on ABA signaling in the moss *Physcomitrella patens*, where ARK (ABA and abiotic stress-responsive Raf-like Kinase), a group B3 MAP kinase kinase (B3-MAPKKK), was identified as an upstream kinase responsible for PpSnRK2s activation [55]. ARK was found to phosphorylate Ser165 and Ser169 of PpSnRK2B. Ser169 corresponds to the serine residues crucial for the activity of Arabidopsis SnRK2s, whereas Ser165 corresponds to Ser171 in SnRK2.6 and Ser154 in SnRK2.10 and NtOSAK, required for the activation in planta. Their results showed convincingly that in vivo PpSnRK2s are activated by ARK phosphorylation and not by autophosphorylation. Recently, four groups have independently reported the activation of Arabidopsis SnRK2s by Raf-like kinases [56,57,58,59]. The Raf-like kinases (48 in Arabidopsis) belong to the MAPKKK family and can be divided into four subfamilies. Those involved in SnRK2s activation belong to subfamily B and appear to act in a stress- and SnRK2 group-specific manner (Figure 2). The role of Raf-like kinases in SnRK2s activation was recently recapitulated in an excellent review by Fàbregas et al. [41]; therefore, only a brief summary will be presented here. Takahashi et al. [57] demonstrated that members of the MAPKK-kinase δ B3 family, M3Kδ1, δ6, and δ7 (aka B3-Raf-like: RAF3, RAF5/AtARK2, and RAF4/AtARK1, respectively) were required for activation of group 3 SnRK2s in ABA and osmotic stress signal transduction. M3Kδ1 phosphorylated in vitro SnRK2s from group 3; in SnRK2.6 Ser171, Ser175, and Thr176 were phosphorylated. As mentioned above, phosphorylation of Ser171 was shown to be required for SnRK2.6 activation by hyperosmotic stress or ABA in planta [44];Takahashi et al. [57] confirmed this. They found that Ser171 was not autophosphorylated but was phosphorylated by Raf-like kinases. Katsuta et al. [59] showed that B3 family MAPKKKs AtARK1/RAF4, AtARK2/RAF5, and AtARK3/RAF6 are upstream activators of group 3 SnRK2s in response to osmotic stress, but to a much lower degree to ABA, and phosphorylate Ser171 and 175 in SnRK2.6. Lin et al. [56] specified that members of subfamilies B2 (RAF10) and B3 (RAF5 and RAF6) activate the ABA-responsive SnRK2s, while RAF24 from the B4 subfamily preferably phosphorylates the ABA-non-responsive kinases. Using multigene *raf* knockout mutants, Lin et al. [60] showed that several kinases of both the B2 and B3 families cooperate in ABA-induced SnRK2s phosphorylation and activation, and that their function is partially redundant, although they seem to exhibit different specificity and most likely participate in different ABA-dependent processes. Soma et al. [58] reported that B4 Raf-like kinases RAF18, RAF20, and RAF24 phosphorylate and activate group 1 SnRK2s in response to osmotic stress in Arabidopsis but do not affect the group 3 (ABA-responsive) ones. Several residues were phosphorylated by RAFs including those corresponding to Ser154 in SnRK2.10, whose phosphorylation is required for further autophosphorylation. This suggested that the phosphorylation by RAFs comprises the first step of SnRK2 activation which is followed by autophosphorylation. Recently, Lin et al. [60] have shown that even though B2 and B3 RAFs phosphorylate both residues mandatory for activation of group 3 SnRK2s, autophosphorylation is responsible for signal amplification by activation of additional SnRK2s molecules. So far, no RAF kinases involved in the activation of group 2 SnRK2s in vivo have been described, albeit in vitro phosphorylation of SnRK2.8 by some B2 and B3 RAFs has been reported [56]. All those results indicate a specificity of Raf-like kinases in respect to SnRK2s activation and the role of phosphorylation of the residue at position −4 in relation to the Ser/Thr crucial for SnRK2 activity. In Arabidopsis, the RAFs engaged in SnRK2 pathways are activated in response to osmotic stress but rather not in response to ABA [60]; their basal activity is sufficient for the phosphorylation of the ABA-dependent SnRK2s. RAFs activity relies on phosphorylation within their activation loop [55,61], with some exceptions [60]. The elements upstream of the RAFs remain unknown.

Several other protein kinases affect SnRK2s activity and will be described later in a chapter dedicated to the crosstalk between ABA and other plant hormones.

## 3. Mechanisms of SnRK2 Inhibition

### 3.1. Dephosphorylation

The inhibition of SnRK2s is mainly achieved by their dephosphorylation and physical interaction with phosphatases (Figure 2). This has been studied mostly for the ABA-responsive kinases. As mentioned earlier, group 3 SnRK2s are inhibited by clade A PP2C phosphatases which together with PYL receptors form the core of the ABA-dependent osmotic stress signaling machinery (for review, see: [17,62]). In the presence of ABA, PYLs bind the hormone and clade A PP2Cs, releasing ABA-dependent SnRK2s from the kinase-phosphatase complex. This prevents kinase dephosphorylation and leads to its activation. The relationship between the PYL ABA receptors and their co-receptors, clade A PP2Cs, has been described in numerous publications and discussed in the comprehensive reviews concerning ABA signaling [42,63,64] and PP2C phosphatases [65]. Therefore, here, we provide only a short overview of PYLs and their role in PP2C inhibition. In Arabidopsis, there are 14 members of the PYL family whose function is partially redundant; they interact with and inhibit clade A PP2Cs in an ABA-dependent manner [46,66,67,68,69]. However, they do show some differences in structure and specificity [70,71,72,73,74,75,76]. Similarly, as PP2Cs and other components of the ABA pathway, PYLs are regulated by various post-translational modifications, mainly phosphorylation and nitration, and by proteasomal degradation, which in turn affect the activity of clade A PP2Cs and thereby of SnRK2s (summarized in [42,63,64,65]). Particularly interesting is phosphorylation of PYLs by the Target of Rapamycin (TOR) kinase enabling a reciprocal regulation of TOR and ABA signaling pathways [1] (Figure 3). Under normal growth conditions, TOR phosphorylates PYLs, which disrupts their association with ABA, restores PP2C activity, and promotes plant growth. Conversely, under stress conditions, ABA-activated SnRK2s phosphorylate the regulatory TOR subunit RaptorB, triggering its dissociation from the TOR complex and inhibition of the TOR activity and eventually leading to the stress response. Recent data indicate that the role of SnRK2s in switching between growth and stress responses is even more complex. Belda-Palazón et al. [2] suggested that the role of ABA-activated SnRK2s in the regulation of TOR signaling is indirect and depends on the formation of a complex with SnRK1. According to their results SnRK2s can promote growth in the absence of ABA via sequestration of SnRK1; the group 3 SnRK2s in complex with PP2Cs interact with SnRK1, and thereby prevent its interaction with TOR allowing plant growth. In the presence of ABA, SnRK2 and SnRK1 kinases are released from the SnRK2-PP2C-SnRK1 complex; SnRK1 interacts with and represses TOR. Even though different pathways of TOR and subsequently growth regulation by SnRK2 have been described, it is very likely that all of them take place in plant cells.

Clade A PP2Cs phosphatases (e.g., ABI1, AHG1, and HAB1) interact with group 3 ABA-responsive SnRK2 kinases and dephosphorylate crucial residues in their activation loop, leading to a significant reduction of the kinase activity [45,77,78]. Soon et al. [79] have shown that the inhibition relies not only on the dephosphorylation but also on the formation of the SnRK2-PP2C complex. One of the PP2C-SnRK2 interaction interfaces is formed by the catalytic sites of the both enzymes, structurally resembling the PYL-ABA-PP2C complex. The second interface, composed of the kinase ABA box and a positively charged surface region of PP2C, stabilizes the complex.

Even though the ABA-box plays an important role in stabilizing the PP2C-SnRK2 interaction, group 1 SnRK2s, which differ significantly within this region from the ABA-responsive ones, can also be regulated by some clade A PP2Cs. Krzywińska et al. [80,81] showed that two PP2Cs (ABI1 and PP2CA) can interact with, dephosphorylate, and effectively inhibit SnRK2.4. This is in line with structural studies, since the key residues involved in the PP2C-SnRK2 interaction are conserved in all SnRK2 groups. However, inhibition of other SnRK2s from group 1 has not been shown so far. As concerns group 2 SnRK2s, the understanding of their regulation is very poor. SnRK2.8 from group 2 was shown to interact with clade A PP2Cs [77,80] and its activity to be regulated by ABI1 and PP2CA in vitro [80,81].

Studies on plant responses to cold led to identification of other PP2Cs inhibiting OST1/SnRK2.6 activity, EGR2 (clade-E Growth-Regulating 2, [82]) and PROTEIN PHOSPHATASE 2C G GROUP 1 (PP2CG1, [83]). SnRK2.6/OST1 is activated upon cold stress in an ABA-independent manner and aids freezing tolerance [32]. Ding et al. [82] have shown that under optimal temperature, EGR2 is *N*-myristoylated by the *N*-myristoyltransferase NMT1. This *N*-myristoylation is required for the localization of EGR2 to the plasma membrane and for enabling its interaction with SnRK2.6/OST1, and subsequently, dephosphorylation and inactivation of the kinase. How the kinase is released from the phosphatase was unclear, since demyristoylating enzymes have not been found in plants. The authors showed that cold stress induces reduced EGR2 *N*-myristoylation and accumulation of de novo-synthesized unmyristyolated EGR2 unable to interact with SnRK2.6/OST1, thereby permitting SnRK2.6/OST1 activation and the stress response. Additionally, the interaction between PP2CG1 and SnRK2.6/OST1 was reduced by cold, and the kinase released from the inhibition. The activated SnRK2.6/OST1 amplified the stress response by PP2CG1 phosphorylation, which suppressed the phosphatase activity [83]. Notably, the inactivation of EGR2 and PP2CG1 is ABA- and PYL-independent.

Another PP2C phosphatase reported as a presumable regulator of ABA-responsive SnRK2s in Arabidopsis is kinase-associated protein phosphatase (KAPP) [84]. It interacts with both SnRK2.2 and SnRK2.3 and down-regulates ABA signaling. However, whether KAPP can dephosphorylate the SnRK2s remains unknown.

Besides PP2Cs, phosphatases from the serine/threonine-specific phosphoprotein phosphatase (PPP) family have also been identified as regulators of SnRK2s. Okadaic acid-sensitive phosphatase(s) were shown to inhibit SnRK2.4 and NtOSAK [7,80]. Type One Protein Phosphatase 1 (TOPP1) was identified as an interacting partner of all ABA-dependent SnRK2s (group 3), SnRK2.8 (group 2), and the ABA-unresponsive SnRK2.4 (group 1) and shown to suppress SnRK2.6 activity [85]. This suppression was enhanced by the TOPP1 regulatory protein AtI-2 (At Inhibitor-2), which promoted the TOPP1-SnRK2 interaction. The inhibition of other SnRK2s by TOPP1 has not been studied. In addition, TOPP1 was found to interact with some PYLs, which led to the inhibition of the phosphatase activity in an ABA-dependent manner. As for PP2Cs, these interactions were also enhanced by ABA. However, TOPP1 exhibited a different pattern of pair-wise interactions with PYLs than PP2Cs, suggesting a preference of certain PYLs to interact with PP2Cs or TOPP1, depending on abiotic stress conditions.

### 3.2. Protein Level

As mentioned earlier, proteasomal degradation of proteins involved in ABA signaling, including SnRK2s, is well documented (for review, see: [42,64,65,86]). Thus, in addition to the inhibition by phosphatases, degradation is believed to be an important mechanism of negative regulation of SnRK2s (Figure 4). The first report that SnRK2s can be degraded by the Ubiquitin Proteasome System (UPS) appeared in 2013, when SnRK2.4 (from group 1) and SnRK2.6 (from group 3) were found in a screen of Arabidopsis proteins undergoing ubiquitination, and whose ubiquitination level increased upon treatment with the proteasomal inhibitor MG132 [87]. Later, it was confirmed that ABA-responsive SnRK2s are indeed degraded by the 26S proteasome. Interestingly, the degradation is kinase-specific. Cheng et al. [88] found that the stability of SnRK2.3, but not SnRK2.2 or SnRK2.6, was affected by PHLOEM PROTEIN 2-B11 (AtPP2-B11), an F-box component of the SKP1 (S PHASE KINASE-ASSOCIATED PROTEIN 1)/Cullin/F-box E3 ubiquitin ligase complex. During seed germination, SnRK2.3 stability was found to be affected by three BTB (Broad-complex, Tramtrack, and Bric-a-brac) proteins—BTB-A2.1, BTBA2.2, and BTB-A2.3—potential substrate adaptors for cullin-based E3-ligases [89]. The stability of SnRK2.6/OST1 (but no other SnRK2) is under control of HOS15 (HIGH OSMOTIC STRESS 15), which is a CULLIN4 (CUL4)-DAMAGED DNA BINDING PROTEIN1 (DDB1)-based E3 ubiquitin ligase [90]. HOS15 binds SnRK2.6/OST1 in its inactive state, i.e., dephosphorylated in the activation loop. In response to ABA, when SnRK2.6 is activated, its interaction with HOS15 is blocked, but later on, when SnRK2.6 becomes dephosphorylated by PP2Cs, it is recognized by HOS15 and targeted to degradation. Expression of *HOS15* and *AtPP2-B11* followed by protein accumulation is induced by ABA, and when their level reaches a threshold, they trigger degradation of SnRK2.3 or SnRK2.6, respectively [88,90]. These results indicate that ABA promotes SnRK2s degradation to inhibit their action at later stages of the ABA response. The degradation can be regulated by SnRK2s phosphorylation by other kinases. Vilela et al. [91] have shown that maize OST1 (ZmOST1) and Arabidopsis SnRK2.6/OST1 are phosphorylated by Casein Kinase 2 (CK2). CK2 phosphorylates conserved serines in the ABA box, which enhances binding with PP2C and thereby OST1/SnRK2.6 dephosphorylation in the activation loop. This allows interaction with HOS15, ubiquitination, and finally, the kinase degradation. Thus, it is very likely that phosphorylation by CK2 facilitates interaction of SnRK2.6 with HOS15, since HOS15 recognizes dephosphorylated SnRK2.6/OST1. Additionally, ubiquitination-mediated degradation can be prevented by Ubiquitin Specific Proteases (UBPs). In *Capsicum annum* CaUBP12 interacts with CaSnRK2.6 and partially inhibits the kinase degradation [92].

Notably, a reciprocal regulation of the UPS by SnRK2s is also possible. It has been shown that Arabidopsis SnRK2.6/OST1 can phosphorylate the RING ZINC-FINGER PROTEIN34 (RZFP34)/CHY ZINC-FINGER AND RING PROTEIN1 (CHYR1, [93]) and PLANT U-BOX (PUB) PUB25 and PUB26 [94] E3 ubiquitin ligases to enhance their activity and positively regulate responses to drought and cold, respectively.

SnRK2s can be degraded not only by UPS. Wang et al. [95] have shown that OST1/SnRK2.6 can be degraded by Senescence-Associated Subtilisin Protease (SASP), and alteration of this process affected plant sensitivity to ABA and drought, but how this process is regulated is unclear.

So far, a specific mode of degradation has been shown for only a few SnRK2s, but it seems likely that the degradation of most, if not all, SnRK2s is specifically regulated to reduce their level when they are not needed. We predict that specific E3 ligases, UPS components, or some proteases controlling the level of other SnRK2s will be identified soon.

It should also be mentioned that expression of several genes encoding SnRK2s is induced in response to various stimuli, especially stresses. *SNRK2* gene upregulation has been described in diverse plants upon treatment with ABA, mannitol, polyethylene glycol, NaCl, water deficit, exposition to extreme temperatures (cold or heat) or some other stressors (for review, see: [8]). It should be noted that even though in most cases an upregulation was observed, repression or no expression changes have also been reported in different plant species. Additionally, besides the up- or down-regulation, *SNRK2* genes differ in the expression kinetics [96]. Unfortunately, only in a few cases in parallel to the analysis of the gene expression was the SnRK2s protein level also monitored. Most likely the enhanced *SNRK2*s expression is a feedback response to the degradation of the protein during progression of the stress response ensuring an appropriate SnRK2 level. Alternatively, an enhanced expression could be needed for the increased SnRK2s synthesis in later stages of the response.

## 4. Regulation of SnRK2 by Secondary Messengers

### 4.1. Calcium Ions

SnRK2s are considered calcium-independent, but there are indications that calcium is in fact involved in their regulation. In a review concerning calcium-regulated protein kinases in plants, Harmon [97] suggested that SnRK2s might be regulated by calcium. This suggestion was based on the properties of the C-terminal domain (domain II) of SnRK2s rich in acidic amino acids potentially involved in Ca^2+^ binding. However, calcium binding by SnRK2s has never been shown directly. Coello et al. [98] reported the existence of a calcium-dependent and ABA-activated SnRK2 in wheat. They showed that calcium enhanced the activity of a partially purified SnRK2 preparation from wheat roots exposed to ABA. Nevertheless, no pure SnRK2 preparation has been shown to be activated by calcium.

On the other hand, several reports have shown an interplay between calcium-binding proteins and SnRK2s. It has been shown that SnRK2s could interact with kinases of the SnRK3 subfamily, also named CALCINEURIN B-LIKE PROTEIN-INTERACTING PROTEIN KINASEs (CIPKs), SALT OVERLY SENSITIVE 2 (SOS2)-LIKE KINASES or PROTEIN KINASES RELATED to SOS2 (PKSs), whose activity depends on calcium binding to their interactor, CALCINEURIN B-LIKE (CBL)/SOS3-LIKE Ca^2+^ SENSOR/BINDING PROTEIN (SCABP) [99,100]. Mogami et al. [101] reported that CIPK26 interacts with SnRK2.2 in vivo and in vitro, and phosphorylates it in vitro. The role of this phosphorylation has not been established. The authors speculated that in vivo the phosphorylation by CIPK26 (and possibly also by other SnRK3s) could modulate SnRK2 activity or the CIPK-SnRK2 interaction and as a consequence affect plant growth under high external Mg^2+^ concentrations.

Finally, there are data showing that calcium can be involved in a negative regulation of SnRK2 activity. Bucholc et al. [102] identified a calcium binding protein, SCS (SnRK2-interacting calcium sensor), which binds to SnRK2s and inhibits their activity. Recent studies have revealed that the use of alternative transcription start sites of the *AtSCS* gene can produce two in-frame transcripts encoding proteins that differ only in their N-terminal fragments. The longer variant, AtSCS-A, contains one canonical EF-hand and three non-canonical EF-hand-like motifs, whereas in the shorter variant, AtSCS-B, the canonical EF-hand is absent. This has major consequences for the mode of SnRK2s inhibition: AtSCS-A requires calcium for the inhibition, while AtSCS-B does not. Interestingly, it has been shown that in vivo AtSCSs preferentially inhibit ABA-responsive SnRK2s [103]. The mechanism of this inhibition has not been fully solved.

Additionally, a recently published discrete dynamic modeling study indicated that PP2Cs (ABI2, AHG3/PP2CA, HAB1) could be inhibited by Ca^2+^. This was confirmed by studying Arabidopsis mesophyll cell lysates, where Ca^2+^ at physiological concentrations inhibited the PP2C activity [104]. Thus, indirectly, Ca^2+^ could have a dual negative (SCS) and positive (PP2Cs) impact on SnRK2 activity, most likely depending on the Ca^2+^ signature.

### 4.2. Reactive Oxygen and Nitrogen Species and Hydrogen Sulfide

There are indications that nitric oxide (NO) is involved in the regulation of SnRK2s and ABA signaling, but whether it acts as a positive or negative regulator is still controversial. In 2006, Lamotte et al. [105] demonstrated that the NO donor DEA/NO (diethylamine NONOate) induced the activation of NtOSAK in *Nicotiana plumbaginifolia*. Then, it was shown that it induced phosphorylation of residues required for the activation—Ser154 and Ser158 [106]. It has also been shown that NO contributes to NtOSAK activation in response to hyperosmotic stress [105,106] or CdCl_2_ [107]; that activation was inhibited substantially by pretreatment of cells with an NO scavenger. Those results suggested that *S*-nitrosylation of NtOSAK could facilitate its phosphorylation and activation. However, in the first minutes of DEA/NO or NaCl treatment, when the kinase becomes active, NtOSAK does not undergo *S*-nitrosylation [106], indicating that the impact of NO on the kinase is indirect, e.g., through modification of other proteins involved in the regulation of the kinase activity.

In the case of ABA-responsive SnRK2s, the role of NO is still disputable. Neill et al. [108] reported that NO is synthetized in response to ABA. Using an NO donor, a scavenger, and a biosynthesis inhibitor, they demonstrated that NO is needed for ABA-induced stomatal closing. This suggests that NO is a positive regulator of ABA signaling and possibly of the SnRK2.6 activity, required for stomatal closure. Similar findings have recently been presented by Li et al. [109]. They showed that glucose-induced stomatal closing is mediated by PYLs, OST1/SnRK2.6, and Slow-type Anion Channel 1 (SLAC1) whose phosphorylation by OST1/SnRK2.6 is needed for stomatal closure, and was positively regulated by Ca^2+^, NO, and reactive oxygen species (ROS). On the other hand, several studies have indicated the opposite. Ribeiro et al. [110] found similar rates of stomatal closing in response to ABA in Arabidopsis wild-type and the double mutant of two nitrate reductases *nia1nia2* with reduced NO production. Lozano-Juste and Leon [111] showed that the *nia1nia2noa1-2* triple mutant is hypersensitive to ABA and more drought-resistant than the wild type. Those results put NO as a negative regulator of ABA signaling and SnRK2 activity, which was confirmed by Wang et al., who showed that NO negatively regulates stomatal closure [112] and ABA-induced arrest of seed germination in Arabidopsis [113] by inhibiting SnRK2.6, SnRK2.2, and SnRK2.3 through *S*-nitrosylation. In SnRK2.6, this modification occurs at Cys137 adjacent to the kinase catalytic site. Since this cysteine is conserved in all SnRK2s, it is plausible that this is a general mechanism of their inhibition. While those results were very convincing, one cannot exclude the possibility that NO could actually have a dual positive and negative impact on SnRK2 activity at different stages of the response. Indeed, a treatment with 50 μM ABA for up to 30 min did not cause SnRK2.6 *S*-nitrosylation, while a longer one, for about 30–60 min, did [112]. Notably, SnRK2.6 became fully active already a few minutes after ABA application. Thus, at the very early stages of the response to stress/ABA, when SnRK2 activity is required, NO could play a positive, likely indirect role, for instance by affecting the activity of PP2Cs or other negative regulators of SnRK2s, while at later stages, when the NO concentration reaches a high level, SnRK2s would become *S*-nitrosylated and thus inhibited. However, the mechanism of the putative positive impact of NO on SnRK2 activity remains purely hypothetical, as no information on such regulation is available.

Several signaling molecules besides NO, e.g., calcium ions, ROS and hydrogen sulfide (H_2_S) are involved in the regulation of stomatal movements [114,115,116]. The production of H_2_S in guard cells can be triggered by ABA [116,117] and ethylene [115]. It has been shown that in guard cells H_2_S in the presence of elevated levels of Ca^2+^ promotes the activation of an S-type anion channel [114], which relies on phosphorylation by SnRK2.6/OST1 [33,34] and calcium-dependent protein kinases [35]. It has been shown that H_2_S enhances SnRK2.6 activity by its persulfidation at Cys131 and Cys137 [118]. Interestingly, Cys137 can also undergo *S*-nitrosylation (see above) in response to ABA. In contrast to the *S*-nitrosylation, persulfidation promotes SnRK2 activity and thereby stomatal closure and drought tolerance, which is in line with their opposed effects on cysteines reactivity. This fits the mode of regulation of SnRK2.6, since it has been shown that H_2_S promotes NO production in response to ABA [116]; thus, most likely, the persulfidation-facilitated activation of SnRK2.6 occurs earlier than its S-nitrosylation. Recently, Chen et al. [53] presented a mechanism of regulation of the SnRK2.6 activity by persulfidation. Persulfidation of Cys131/137 changes the structure of SnRK2.6; Ser175 residue comes in close proximity to Asp140, the ATP-γ-phosphate proton acceptor site, which results in more efficient Ser175 autophosphorylation, followed by enhanced autophosphorylation of several other residues and interaction of the kinase with its targets. Additionally, they showed that phosphorylation of Ser267 has a positive impact on persulfidation.

SnRK2s are also affected by ROS; however, as in the case of NO, the mechanism has not been fully elucidated. Kulik et al. [107] have shown that the ABA-non-responsive NtOSAK is transiently activated in tobacco BY-2 cells in response to H_2_O_2_ treatment. In contrast, an exposure of *Brassica napus* BnSnRK2.4-1C, closely related to Arabidopsis SnRK2.4 and SnRK2.10 and tobacco NtOSAK, to H_2_O_2_ in vitro inhibited its autophosphorylation and kinase activity [119]. The activity could be restored and even enhanced by incubation with reductants, proving that the kinase was modulated by a redox mechanism. The inhibition was connected with oxidation of Cys233. A transient inhibition of BnSnRK2.4-1C was also observed in vivo, in *B. napus* seedlings exposed to H_2_O_2_. Although the above results obtained for different experimental models differed, in both cases, the regulation was transient. In the case of NtOSAK, the effect of H_2_O_2_ has not been analyzed in vitro. In vivo, the kinase activity is affected not only by its direct modification but also indirectly, via regulation of other proteins having an impact on the kinase. H_2_O_2_ inhibits PP2C phosphatases, major regulators of SnRK2 activity, as shown for ABI1 [120], ABI2 [121], and HAB1 [122]. H_2_O_2_ caused a reversible oxidation of HAB1 at Cys186 and Cys274, which inhibited its catalytic activity and prevented the interaction with SnRK2.6. As with BnSnRK2.4-1C, HAB1 could be reactivated with reducing agents. The authors suggested that H_2_O_2_, which appears early in the response to stress, inactivates HAB1 and other PP2Cs to allow SnRK2 activation. It should be added that in response to ABA or some stressors, SnRK2s are involved in the regulation of ROS homeostasis [107,123,124,125]. It is quite likely that the reversible oxidation of SnRK2s and PP2Cs could constitute a feedback mechanism, allowing fine-tuning of the response to environmental and developmental changes. The regulation of SnRK2 activity, ABA signaling, and stress responses in general by NO, H_2_S, or ROS seems to be highly dependent on the stage of the response.

### 4.3. Phosphatidic Acid

The ABA-independent kinases SnRK2.4 and SnRK2.10 have been identified among proteins interacting with phosphatidic acid (PA, [126]) and, unlike SnRK2.6, binding to liposomes containing PA [24]. Therefore, it was highly likely that PA, whose role in stress signal transduction is well documented, acts as a regulator of at least some SnRK2s. In support, Klimecka et al. [127] have shown that PA modulates SnRK2.4 structure and phosphorylation of some SnRK2 targets. Moreover, similarly as ROS, PA inhibits the activity of ABI1 and PP2CA/AHG3 phosphatases [127,128], suggesting that, in planta, it could be an important regulator of group 1 SnRK2s. PA affects not only the activity but also localization of its targets. The PA binding is responsible for membrane association of SnRK2.4 and SnRK2.10 [21,24], as well as some of the SnRK2 regulators such as ABI1, PP2CA, or SCS [127,128].

## 5. Impact of Plant Hormones Other Than Abscisic Acid on SnRK2 Activity

There is a complex crosstalk between plant hormones. In general, ABA functions antagonistically to the hormones promoting cell proliferation and growth, such as gibberellins (GAs), brassinosteroids (BRs), cytokinins, and auxins.

Several proteins link the BR and ABA pathways (Figure 5), among them, the negative regulator of BRs signaling BR-Insensitive 2 (BIN2, [129,130,131]). BIN2 is a GSK3-like kinase [132] which phosphorylates SnRK2.2 and SnRK2.3 [133]. In SnRK2.3 Thr180 is the main residue phosphorylated by BIN2 and its phosphorylation enhances the SnRK2 activity. Wang et al. [134] have shown that, similarly to SnRK2s, BIN2 is dephosphorylated and inhibited by the ABI1 and ABI2 phosphatases, while ABA prevents this dephosphorylation by inhibiting the phosphatases in a PYL-dependent manner. Another BR-related protein regulating SnRK2 activity is the BR co-receptor BRASSINOSTEROID INSENSITIVE1-ASSOCIATED RECEPTOR KINASE 1 (BAK1), which phosphorylates OST1/SnRK2.6 to stimulate its activity and ABA-induced stomatal closure [45,135]. On the other hand, BRs inhibit the formation of the BAK1/SnRK2.6 complex and the phosphorylation of SnRK2.6, thereby countering the action of ABA.

Similarly, ABA and cytokinin signaling in the regulation of plant development and abiotic stress responses affect each other. Huang et al. [136] described the molecular mechanism of reciprocal regulation of ABA and cytokinin signaling relying on interactions of Arabidopsis response regulators (ARRs) with ABA-responsive SnRK2s. Type-B ARRs (ARR1, ARR11, and ARR12), which are positive transcriptional regulators of cytokinin signaling, have been shown to inhibit the activity of SnRK2.6 and subsequently ABA-dependent control of germination. On the other hand, SnRK2.2, SnRK2.3 and SnRK2.6 phosphorylate and stabilize type A ARR (ARR5), which is a negative regulator of cytokinin signaling.

The GA and ABA signaling pathways are also linked. Lin et al. [137] found that in rice Tiller Enhancer (TE), an activator of the E3 ubiquitin ligase complex APC/C^TE^, facilitates degradation of OsPYL/RCAR2, OsPYL/RCAR9 and OsPYL/RCAR10. GA enhances the TE—PYL interaction promoting PYL degradation, thus reducing the activity and also protein level of group 3 SnRK2s: SAPK8, SAPK9, and SAPK10. However, in the presence of ABA SnRK2s phosphorylate TE, which releases PYLs from the complex with TE and prevents their degradation.

## 6. ABA-Dependent versus ABA-Independent Regulation of SnRK2s

As has been summarized in previous chapters, the ABA-responsive kinases can be regulated at several different levels. Presumably, this is also true for the ABA-non-responsive ones; however, in contrast to the fairly well studied group 3 SnRK2s, those from the two other groups remain largely neglected. The fact that the mode of regulation of group 3 SnRK2s differs from that of group 1 kinases has been known since 2010, when Vlad et al. [44] showed that the mechanisms of phosphorylation of ABA-responsive (SnRK2.6) and ABA-unresponsive (SnRK2.10) SnRK2s differ. Phosphorylation of Ser171 and Ser175 in SnRK2.6 occurs independently in response to osmotic stress or ABA, while in SnRK2.10, this process is sequential: phosphorylation of Ser154 is required for the modification of Ser158. In addition, group 3 SnRK2s are phosphorylated at a basal level in non-stress conditions and their phosphorylation increases upon stress, while the phosphorylation of group 1 SnRK2s is undetectable in control conditions and appears rapidly upon osmotic stress [23,43,44], additionally indicating differences in regulation between the ABA-responsive and non-responsive SnRK2s. Recently, SnRK2s from group 1 and 3 have been shown to be phosphorylated by RAF-like kinases, albeit by different ones. Group 1 SnRK2s were phosphorylated by RAFs from group B4 in response to osmotic stress [56,58], whereas group 3 SnRK2s were phosphorylated by B2 and B3 family members in response to osmotic stress or ABA [56,57]. Importantly, the activation of group 1 and group 3 SnRK2s in response to osmotic stress was abolished in appropriate *raf* mutants, while the response of group 3 kinases to ABA was only partially reduced [59], suggesting that autophosphorylation or phosphorylation by upstream kinase(s) other than the RAF ones is needed for their full activation in response to ABA.

While the activation of the ABA-non-responsive SnRK2s is only partially understood, even less is known about the opposite process. It is well established that phosphoprotein phosphatases negatively regulate SnRK2s, both ABA-responsive and non-responsive ones, but among group 1 SnRK2s, only one, SnRK2.4, has been shown to be inactivated by clade A PP2Cs (ABI1 and PP2CA) and okadaic acid-sensitive PPPs, while for group 2 only an in vitro inhibition of SnRK2.8 by some PP2Cs has been documented [80,81]. Moreover, the mechanism of inhibition of the phosphatases that would allow SnRK2s phosphorylation and activation in an ABA-independent manner remains largely unknown. Some reports suggest that the basal PP2C activity could be controlled by interaction with monomeric PYL receptors also in an ABA-independent manner [70]. It has been shown that in the absence of ABA, PYL13 preferentially inhibits PP2CA and interacts with other PYLs, and the interaction antagonizes their activity [73,74]. However, others have reported opposite results showing that PYL13 inhibits PP2CA (as well as ABI1 and ABI2) only in the presence of ABA [138]. Moreover, there are no data indicating that the interaction of PP2CA or ABI1 with some PYLs could be responsible for releasing group 1 SnRK2s from the PP2C inhibition in response to osmotic stress. On the contrary, it is very likely that PYLs are negative regulators of their activity. In a *pyl1/2/3/4/5/7/8/9/10/11/12* knockout mutant, the activation of group 3 SnRK2s in response to ABA was completely blocked but the activation of the same SnRK2s by osmotic stress was enhanced [75], revealing the opposite regulation of SnRK2s by PYLs in the ABA-dependent and ABA-independent pathways.

Finally, SnRK2s as well as their regulators are controlled by degradation. While the specificity and timing of SnRK2s degradation in ABA signaling is fairly well described, nearly nothing is known about such regulation in the case of ABA-non-responsive SnRK2s. Nevertheless, one can expect UPS degradation to be involved in the regulation of ABA-non-responsive SnRK2s as well, especially since they are activated mainly at the early stages of the stress response.

## 7. Concluding Remarks and Future Perspectives

Recent years have provided ample data on the regulation of ABA-responsive SnRK2s and ABA signaling by numerous mechanisms including phosphorylation, oxidation, *S*-nitrosylation, persulfidation, diverse protein–protein interactions, and finally, degradation. However, our understanding of this vast subject is still merely a tip of an iceberg. Numerous gaps and questions remain that need to be addressed, for example, those regarding the role of secondary messengers and posttranslational modifications (PTMs) other than phosphorylation within the kinase activation loop and their mutual interplay. This is a complex issue since SnRK2s regulate various processes, both normal plant growth and responses to diverse environmental stresses and, as we now know, in several cases, the SnRK2s’ regulatory elements/modifications (or their functions) are stress-specific. It is very likely that depending on the stage of plant development, type of stress, its amplitude and duration, the role of given PTM or secondary messengers could differ; for instance, NO, ROS, or Ca^2+^ could be involved, directly or indirectly, in both activation and inactivation of SnRK2s. We predict that a systematic analysis of the PTMs of SnRK2s and their regulators will provide important data concerning SnRK2s regulation and the fine tuning of various signaling pathways they are involved in.

The possible regulation of SnRK2s activity by their subcellular localization, which seems to be modulated in a stress-specific manner, is still not explored. It is unclear whether SnRK2 relocalization is merely a result of its activation and interaction with substrates with a specific cellular location, or is a means of regulation. The localization can determine the interaction partners of the kinase and can also lead to either activation (by separating it from a negative regulator) or inhibition. How the localization is related to the kinase activity remains to be determined.

Not all these issues can be addressed at once. In our opinion, in the near future, attention should be focused on the ABA-non-responsive SnRK2s from groups 1 and 2 and the ABA-independent regulation of all SnRK2s. We still do not know the mechanisms responsible for releasing SnRK2s from their inhibitors in an ABA-independent manner. Are there any proteins involved in this process or does it depend only on PTMs? How do PYLs control the ABA-independent activation of SnRK2s? Is there a crosstalk between the ABA-dependent and non-dependent kinases and how does it impact their activity?

Another problem requiring more attention is the degradation of the ABA-non-responsive SnRK2s, as so far there is no information on the ubiquitin E3 ligases or proteases which could control the level of these kinases in plant cells.

To conclude, despite the constant progress in the understanding of the mechanisms of SnRK2s regulation, we are very far from having a full picture of this complex, multifaceted process.

## Figures and Tables

**Figure 1 cells-10-02180-f001:**
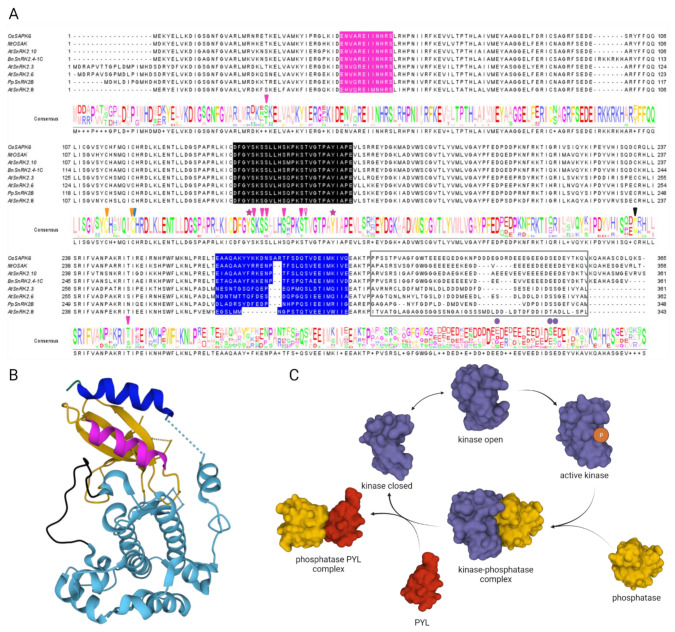
Sequence conservation, structural features, and conformational dynamics of SnRK2s. (**A**) Alignment of selected SnRK2s amino acid sequences with structural features and post-translational modifications (PTMs) indicated. αC helix in magenta, activation loop in black, and SnRK2 box (domain I) in blue. Domain II is boxed. PTMs marked as follows: solid magenta arrowheads indicate Ser residues phosphorylated by autophosphorylation and Raf-like kinases (RAFs), empty magenta arrowhead indicates Thr residue phosphorylated by RAFs and Brassinosteroid Insensitive 2 kinase (BIN2), solid magenta stars indicate phosphorylated Tyr residues, solid blue arrowhead indicates Cys residue undergoing S-nitrosylation, solid orange arrowheads indicate Cys residues undergoing persulfidation, solid black arrowhead indicates Cys modified by oxidation, and purple dots indicate Ser phosphorylation by Casein Kinase 2 (CK2). (**B**) Crystal structure of SnRK2.6 (PDB 3UC4). Coloring of structural features as in (**A**). Additionally, N-lobe and C-lobe are colored in yellow and light blue, respectively. (**C**) Basic model of ABA-dependent SnRK2 self-activation mechanism. SnRK2 regulation is based on reversible protein phosphorylation. When abscisic acid (ABA) level is low clade A type 2C protein phosphatases (PP2Cs) inactivate SnRK2s due to direct interaction and dephosphorylation. When ABA level increases, ABA receptors PYRABACTIN RESISTANCE 1/PYR1-LIKE/REGULATORY COMPONENTS OF ABA RECEPTOR (PYLs) bind ABA and interact with PP2Cs thus SnRK2s can be released from the kinase-phosphatase complex. Once released, SnRK2s can spontaneously switch between open (inactive) and closed (active) conformation and become activated due to phosphorylation in the SnRK2 kinase activation loop.

**Figure 2 cells-10-02180-f002:**
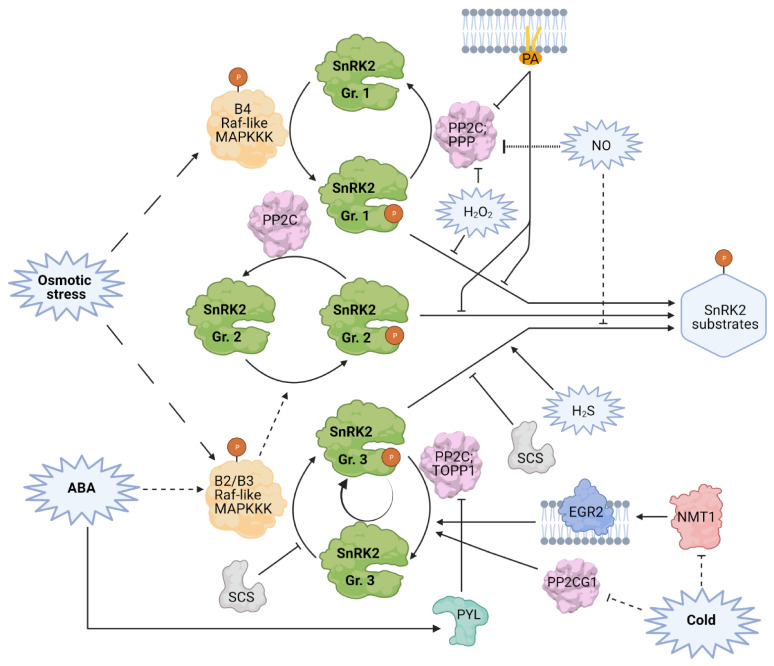
Diverse modes of regulation of SnRK2 activity. Solid lines indicate experimentally shown direct interactions, dashed lines indicate indirect interaction or multistep process. Dotted lines represent hypothetical regulation. Lines with arrowheads indicate positive regulation, bar-ended lines indicate inhibition. SnRK2s activation relies on phosphorylation of specific residues in their activation loops. All SnRK2s are activated by osmotic stress, in the case of SnRK2s from group 1 and 3 via phosphorylation by Raf-like MAPKKK from subfamilies B4 or B2 and B3, respectively. SnRK2 kinases from group 3 and (most likely) 2 are activated by ABA via phosphorylation by Raf-like MAPKKK from subfamilies B2 and B3. ABA-dependent activation can be amplified by autophosphorylation. In the absence of ABA, SnRK2s from group 3 interact with type 2C protein phosphatases (PP2Cs) and Type One Protein Phosphatase 1 (TOPP), which dephosphorylate their activation loops, rendering them inactive. When present, ABA binds to PYL (PYR1-LIKE) ABA receptors, which causes their interaction with and inactivation of the phosphatases, allowing for SnRK2 release and activation. Other PP2Cs—clade E Growth-Regulating 2 (EGR2) and PROTEIN PHOSPHATASE 2C G GROUP 1 (PP2CG1)—negatively regulate group 3 SnRK2 kinases. Cold attenuates SnRK2 inhibition by the latter two, in the case of EGR2 by reducing its *N*-myristoylation. SnRK2-interacting calcium sensor (SCS) negatively impacts the function of group 3 SnRK2s by blocking kinase activation and activity towards substrates. SnRK2s from all groups are negatively regulated by PP2Cs and serine/threonine-specific phosphoprotein phosphatases (PPPs). Secondary messengers—PA (phosphatidic acid), H_2_S, NO, H_2_O_2_—impact the activity of specific SnRK2s. H_2_S enhances SnRK2.6 activity by persulfidation of its Cys residues. NO inhibits group 3 SnRK2s through S-nitrosylation; on the other hand, it contributes to activation of group 1 SnRK2, probably in an indirect fashion. H_2_O_2_ inhibits the activity of BnSnRK2.4-1C from group 1 via oxidation of Cys233. PA, H_2_O_2_, and probably NO also affect the activity of PP2Cs in their regulation of group 1 SnRK2s. Therefore, those secondary messengers can have a dual, positive, and negative role in the regulation of SnRK2s.

**Figure 3 cells-10-02180-f003:**
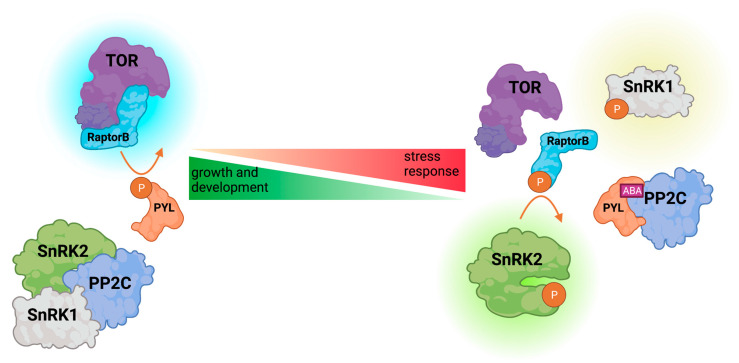
Schematic illustration of the TOR-SnRK-dependent growth/stress response balance regulation in Arabidopsis. Under favorable conditions, active TOR (Target of Rapamycin) kinase complex phosphorylates PYL, compromising its interaction with PP2C phosphatase, leading to SnRK inactivation. Upon stress, abscisic acid (ABA) enables PYL-PP2C interaction, thus releasing SnRK from inhibition. Active SnRK phosphorylates RaptorB, rendering TOR inactive. Active kinases are indicated by a bright glow.

**Figure 4 cells-10-02180-f004:**
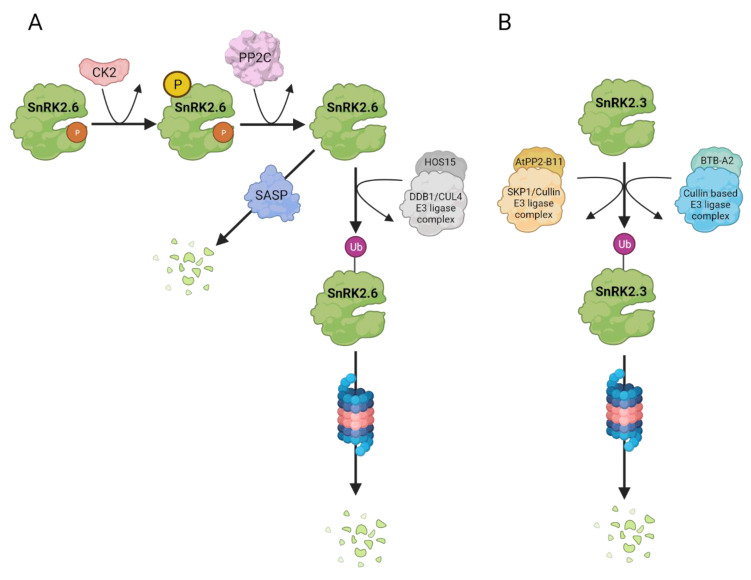
Controlled degradation of SnRK2s in Arabidopsis. ABA-responsive SnRK2s undergo degradation in a kinase-specific manner via proteasomal degradation pathway as well as by specific proteases. (**A**) SnRK2.6 ubiquitination occurs when the kinase is in an inactive, dephosphorylated state. The kinase interacts with HIGH OSMOTIC STRESS 15 (HOS15), a CULLIN4 (CUL4)-DAMAGED DNA BINDING PROTEIN1 (DDB1)-based E3 ubiquitin ligase and is targeted to degradation via Ubiquitin Proteasome System (UPS). Phosphorylation of SnRK2.6 within the ABA-box, catalysed by Casein Kinase 2 (CK2), promotes SnRK2.6 degradation by enhancing its interaction with and dephosphorylation by clade A PP2Cs, and thus interaction of SnRK2.6 with HOS15. Additionally, SnRK2.6 protein level is regulated via Senescence-Associated Subtilisin Protease (SASP). (**B**) SnRK2.3 stability is negatively controlled by PHLOEM PROTEIN 2-B11 (AtPP2-B11), an F-box component of the SKP1 (S PHASE KINASE-ASSOCIATED PROTEIN 1)/Cullin/F-box E3 ubiquitin ligase complex as well as Broad-complex, Tramtrack, and Bric-a-brac proteins, namely BTB-A2.1, BTBA2.2, and BTB-A2.3, substrate adaptors for cullin-based E3-ligases. The ubiquitinated kinase is degradated by UPS.

**Figure 5 cells-10-02180-f005:**
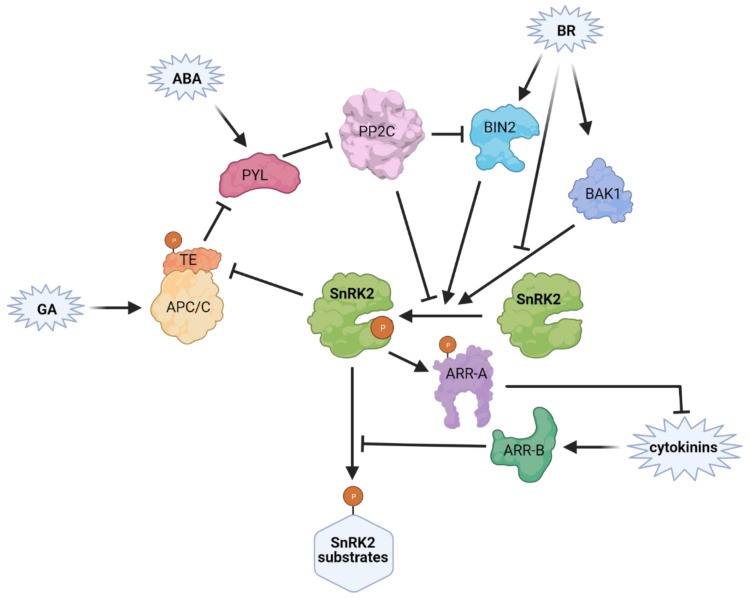
Plant hormones in SnRK2 regulation. Lines with arrowheads indicate positive regulation, bar-ended lines indicate inhibition. SnRK2 activity is impacted by abscisic acid (ABA), brassinosteroid (BR), giberellin (GA) and cytokinin signalling pathways and crosstalk between those hormones. BRASSINOSTEROID INSENSITIVE 2 kinase (BIN2), a negative regulator of BRs signaling, phosphotylates SnRK2.2 and 2.3 and enhances their activity. In the absence of ABA, group 3 SnRK2s and BIN2 are inhibited by clade A type 2C protein phosphatases (PP2Cs). In the presence of ABA, the hormone binds to PYRABACTIN RESISTANCE 1/PYR1-LIKE/REGULATORY COMPONENTS OF ABA RECEPTOR (PYLs) which interact with and inhibit PP2Cs. BR co-receptor BRASSINOSTEROID INSENSITIVE1-ASSOCIATED RECEPTOR KINASE 1 (BAK1) through SnRK2.6 phosphorylation induces ABA-dependent stomatal closure, whereas BRs negatively regulate BAK1/SnRK2.6 complex formation. Positive regulators of cytokinin signaling, Type-B Arabidopsis response regulators (ARRs; namely ARR1, ARR11, and ARR12) interact with and inhibit SnRK2.6 activity. On the other hand, a negative regulator of cytokinin signalling, Type-A ARR (ARR5), is stabilized through phosphorylation catalysed by SnRK2s. The link between GA and ABA pathways relies on interaction of an activator of the E3 ubiquitin ligase complex APC/C^TE^—Tiller Enhancer (TE) with PYL. GA enhances the interaction, which leads to PYL degradation and reduction of the activity and protein level of group 3 SnRK2s. The presence of ABA induces TE phosphorylation by SnRK2s from group 3 and PYL dissociation from TE-PYL complex, thus preventing the GA-induced degradation.

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
