# Peer review of "The Multifaceted Regulation of SnRK2 Kinases"

_cells, 2021, doi:10.3390/cells10092180_

Round 1
Reviewer 1 Report
In the review article entitled "The Multifaceted Regulation of SnRK2 Kinases", the authors reviewed the regulation of SnRK2s in plant cells. The manuscript is timely and well prepared, and the figure is well drawn. Before publishing, some relevant description need to be added in the manuscript.
Some comments:
- Some detailed description about posttranslational modifications of SnRK2s need to be added in “4.2. Reactive Oxygen and Nitrogen Species and Hydrogen Sulfide”. For example, the persulfidation of Cys131/137 altered SnRK2.6 structure, resulted in serine Ser175 residue more close to aspartic acid Asp140, who belong to ATP-γ-phosphate proton acceptor may effectively improve the transfer efficiency of phosphate groups to Ser175, thus persulfidation enhanced the phosphorylation level of Ser175.
- In line 475, the author needs to briefly introduce the direct relationship between signaling molecules(ABA, H2S and Ca2+) and 6 in guard cells. Please refer to and cite relevant literature.
Reference:
Wang L, Wan R, Shi Y, Xue S. Hydrogen Sulfide Activates S-Type Anion Channel via OST1 and Ca(2+) Modules. Mol Plant. 2016 7;9(3):489-491. https://doi: 10.1016/j.molp.2015.11.010.
Jia, H., Chen, S., Liu, D., Liesche, J. et al Ethylene-induced hydrogen sulfide negatively regulates ethylene biosynthesis by persulfidation of ACO in tomato under osmotic stress. Front. Plant Sci. 2018 9:1517. https://doi.org/10.3389/fpls.2018.01517
Reviewer 2 Report
This is a well written and informative review addressing a very important theme – regulation of SnRK2 kinases involved in the control plant development and adaption to changing environment. Authors competently analyzed a lot of data accumulated recently on the theme paying attention to both ABA-dependent and independent mechanisms of regulation. I attentively and with great interest read the article. Still I have some suggestions that may hopefully make it easier for readers to understand rather complicated regulatory networks described in the article.
- Figure legends should be supplied with short description of the schemes presented on them. Figures should be self sufficient and understandable without necessity to search for the explanations and deciphering of abbreviations in the text. This concerns all the figures, but figure 1c, first of all.
- In 3.1 section, description of the involvement of PYLs in control of SnRK2s dephosphorylation and their activity is rather unclear. Although it is described above, here it is also necessary to mention that ABA prevents SnRK2s dephosphorylation by inhibiting phosphatases in a PYL-dependent manner.
- “Particularly interesting is phosphorylation of PYLs by the Target of Rapamycin (TOR) kinase enabling a reciprocal regulation of TOR and ABA signaling pathways [1]. Under normal growth conditions TOR phosphorylates PYLs, which disrupts their association with ABA, restores PP2C activity and promotes plant growth. Conversely, under stress conditions ABA-activated SnRK2s phosphorylate the regulatory TOR subunit RaptorB triggering its dissociation from the TOR complex and inhibition of the TOR activity and eventually leading to the stress response. “
- I think this is rather important fragment illustrating an important phrase in the abstract telling that “The SnRK2s can be viewed as molecular switches between growth and stress response”. I advise to add one more figure with a scheme illustrating this switching mechanism.
- “whose ubiquitination level increased upon MG132 treatment”
– abbreviation MG132 is not deciphered and mechanism of its action is unclear. I think it may be mentioned that MG132 is proteasome inhibitor. In the cited article it is said that “The ubiquitylation state of more than half of the targets increased after treating seedlings with the proteasome inhibitor MG132”.
- In the section concerning Calcium ions it is said that “CIPK26 (CALCINEURIN B-LIKE PROTEIN-INTERACTING PROTEIN KINASE 26), a member of the SnRK3 subfamily whose activity depends on calcium, interacts with SnRK2.2”
- It seems to me that this information may look more intriguing if it will be specified that the calcineurin B-like proteins are alternatively designated as SOS3-like (salt overly sensitive) involved in salt-stress responses.
I have just several more minor remarks
- “SnRK2s together with several other signaling proteins, including kinases, play a key role”
– I think that this sentence should be rephrased. It is present form is sounds as if SnRK2s are not kinases themselves. May not it be “including OTHER kinases”?
- “Beside PP2Cs also phosphatases from the serine/threonine-specific phosphoprotein phosphatase (PPP) family have been identified as regulators of SnRK2s.”
- It seems to me that position of “also” should be changed: “Beside PP2Cs, phosphatases from the serine/threonine-specific phosphoprotein phosphatase (PPP) family have also been identified as regulators of SnRK2s.
- “Unfortunately, only in a few cases in parallel to the analysis of the gene expression also the SnRK2s protein level was monitored”
- Again I am not happy with position of “also”. I propose “Unfortunately, only in a few cases in parallel to the analysis of the gene expression the SnRK2s protein level was also monitored”
Round 2
Reviewer 1 Report
Agreed to publish
Reviewer 2 Report
Authors carefully followed all my recommendations and I think it may be published in tis present form